# Study on the Effect of Man-Machine Response Mode to Relieve Driving Fatigue Based on EEG and EOG

**DOI:** 10.3390/s19224883

**Published:** 2019-11-08

**Authors:** Fuwang Wang, Qing Xu, Rongrong Fu

**Affiliations:** 1School of Mechanic Engineering, Northeast Electric Power University, Jilin 132012, China; 2201800279@neepu.edu.cn; 2College of Electrical Engineering, Yanshan University, Qinhuangdao 066004, China; frr1102@ysu.edu.cn

**Keywords:** driving fatigue relieving, MRM, brain network, eye movement, SQ

## Abstract

Rapid and accurate detection of driver fatigue is of great significance to improve traffic safety. In the present work, we propose the man-machine response mode (MRM) to relieve driver fatigue caused by long-term driving. In this paper, the characteristics of the complex brain network, which can effectively reflect brain activity information, were used to detect the change of driving fatigue over time. Combined with the traditional eye movement characteristics and a subjective questionnaire (SQ), the changes in driving fatigue characteristics were comprehensively analyzed. The results show that driving fatigue can be effectively delayed using the MRM. Additionally, the response equipment is low in cost and practical, so it will be practical to use in actual driving situations in the future.

## 1. Introduction

Fatigue driving is one of the main causes of traffic accidents [1,2]. Previous investigations found that 15–20% of fatal crashes involve driver fatigue [3,4,5]. Therefore, it is necessary to quickly and accurately detect driving fatigue and take measures to relieve it in time. Researchers have mainly studied the problem of driving fatigue detection from subjective and objective aspects. The former type of judgment of driving fatigue is mainly based on the subjective judgment of the driver [6]. The latter mainly judges driving fatigue based on physiological characteristics of drivers. Previous studies have found that human physiological signals, such as electrocardiogram (ECG), electrooculogram (EOG) and electroencephalogram (EEG), can effectively reflect people’s mental fatigue.

There have been many studies on mental fatigue based on human physiological signal characteristics [7,8]. Previous studies have showed that human mental fatigue is associated with eye movement characteristics [9,10,11]. Di Stasi et al. found that the saccadic eye movement parameters are sensitive indicators for human mental fatigue [12]. Schleicher et al. found a strong correlation between human mental fatigue and eye movement [13]. Cruz et al. showed that the eye and eyelid movement can be used as predictors of performance decrement resulting from mental fatigue [14]. There have been many studies on human mental fatigue based on EEG signal features. Four frequency sub-bands δ (0–4 Hz), θ (4–8 Hz), α (8–14 Hz) and β (14–32 Hz) are widely applied to analyze the state of driving fatigue. For example, the relative energy ratio (α + β)/δ1 [15] is an algorithm that uses the EEG alpha spindle and detection of the alpha band relative energy to signify the level of driving fatigue [5]. A previous study has shown that the brain fatigue characteristics can be easily detected from the posterior (P3, P4) brain region using EEG signals [7]. Therefore, the leads associated with the brain region can be used as the preferred ones for the analysis of driving fatigue in this study. In our study, we mainly measure the driving fatigue based on the characteristics of EEG and EOG.

There are many traditional ways to relieve human mental fatigue, such as reducing the intensity of work, having a short rest, drinking caffeinated beverages and taking special medicines. In recent years, electrical stimulation has been widely studied as a novel method to relieve human fatigue [16,17]. Dailey et al. found that human pain and fatigue can be relieved by using transcutaneous electrical nerve stimulation [18]. Zhao concluded that human physiological fatigue can be relieved by stimulating points of the human body [19]. In terms of human cognition, some studies have shown that a proper number of cognitive tasks are more conducive to alleviating mental fatigue than a small number. Oron-Gilad et al. demonstrated that some cognitive tasks, especially ones involving understanding and memory, could keep drivers alert [20]. Research by Verwey and Zaidel showed that when drivers perform an attention-demanding secondary task while driving, this can effectively improve alertness [21]. Drory’s experimental results showed that a non-mandatory driving-related task of a reasonable size could make a driver maintain high levels of alertness [22]. Gershon et al. indicated that undertaking an interactive cognitive task could effectively inhibit driving fatigue [23]. In our study, based on the above research [20,21,22,23], we adopted a method to appropriately increase the driver’s cognitive intensity (for example, by doing a non-mandatory driving-related task) to relieve driving fatigue caused by long-term driving.

## 2. Materials and Methods

### 2.1. Subjects

A total of 12 healthy subjects (10 males and 2 females; aged 32 ± 1.6 (standard deviation, SD)) were selected from a pool of volunteers to carry out the experiment. All subjects had to hold a driver’s license. Additionally, they needed to satisfy the conditions of having no sleep diseases and no other mental disease. They also had to be free of medication during the experiment. During the experiment, subjects were not allowed to drink alcohol, coffee or other nerve-stimulating drinks. Additionally, the experiment was divided into two types, with the normal driving mode being arranged in the first type of experiment and the man-machine response mode (MRM) being arranged in the second one. All subjects needed to complete the two types of driving mode experiments. It should be noted that the MRM mode of experiment can only be carried out after all subjects have completed the first type of experiment. All subjects were numbered from 1 to 12. In the experiment, the subjects were tested in sequence according to their number. Only one subject was assigned to the experiment each day.

### 2.2. Procedure and Electroencephalogram (EEG) Recording

This experiment was carried out in a laboratory-simulated driving environment. The indoor temperature of the laboratory was 20–25 °C, and the humidity was 30–50% relative humidity (RH). A vehicle simulator (JT/T378) was used to simulate the real driving environment. This model of the simulator included several main components such as the accelerator, clutch, foot brake, computer, steering wheel, hand brake, and liquid crystal display. The upper computer driving software included a variety of driving modes, one of which was the driving teaching mode. In the experiment stages, all subjects were required to drive continuously at a speed of 80–100 km/h. The weather was sunny, with broad vision and good visibility. Meanwhile, the subjects were asked to drive in a road environment with low traffic volume. All subjects were instructed to drive in automatic mode. Movements unrelated to the experiment, such as head and hand movements, were avoided as much as possible to reduce artefacts in the EEG recording.

This experiment was carried out using a car simulator. Figure 1 shows the experimental environment and the experimental equipment used. All subjects continuously drove for three hours (13:00-16:00 p.m.). Seven sets of data were collected for every subject. The data acquisition was divided into seven stages (stage 1-13:00 p.m., stage 2-13:30 p.m., stage 3-14:00 p.m., stage 4-14:30 p.m., stage 5-15:00 p.m., stage 6-15:30 p.m. and stage 7-16:00 p.m.). For each stage, data collection lasted for 3 min at a time. Additionally, half an hour of sleep (12:00–12:30 p.m.) was arranged for all subjects to avoid the influence of fatigue due to a lack of sleep. We collected EEG data using Neuroscan, whose electrodes (Ag/AgCl) were attached to the scalp according to the international 10–10 system.

Previous studies have shown that EEG equipment with fewer electrodes (Emotiv) can effectively monitor people’s driving fatigue [24,25,26,27]. The Emotiv EEG acquisition equipment has 14 electrodes (14 channels = AF3, AF4, F3, F4, FC5, FC6, F7, F8, T7, T8, P7, P8, O1, and O2), which is consistent with the 14 electrodes (14 channels = F3, F4, F7, F8, FT7, FT8, C3, C4, TP7, TP8, P3, P4, O1, and O2) on the Neuroscan device selected in this paper. The sampling rate of this device is selected to be 1000 Hz. In addition, the 14 electrodes of these two devices are placed in almost the same positions on the surface of the brain’s cerebral cortex. Therefore, in this paper, we chose a relatively small number of 14 electrodes to study driving fatigue to effectively reflect the brain activity features in the frontal, central, and posterior regions of the human brain.

In our study, all subjects were informed of the research background and research plan of the experiment. Additionally, all subjects were free to choose whether to continue participating in the experiment or not. Moreover, all of them gave their written informed consent to be included in the study. The Ethics Committee at the Northeast Electric Power University Hospital endorsed the study protocol, according to the Code of Ethics of the World Medical Association (Declaration of Helsinki).

The experiments were carried out in a laboratory simulation environment. Figure 1 shows the experimental set-up.

In the experiment, the MRM was used to relieve driver fatigue caused by long-term driving. The hardware of the MRM mainly includes an answer switch, a Micro Control Unit (MCU), a speaker, and a voice chip module (ISD1700) produced by the Information Storage Devices. For the MCU, the model we chosen was the STM32F103RCT6. There are three ways in which fatigue can be relieved by adopting this mode. Firstly, the MCU randomly gives two two-digit numbers through the speaker, which take values from 0 to 100. After hearing these two numbers, the subjects need to calculate the sum of the two numbers and judge whether the sum is three digits. If this sum exceeds three digits, the subjects press the “right” answer switch, otherwise press the “left” answer switch. Secondly, the subjects need to judge whether this sum is divisible by 3. Similarly, subjects need to press the “right” answer switch on the steering wheel if the sum can be divided by 3, otherwise press the “left” answer switch on the steering wheel. The subjects need to complete both responses within 5 s, otherwise, it is regarded as a mistake. Thirdly, the MCU plays cheerful music (Bandari Light Music) for 3 s through the speaker if the response is correct, otherwise, the speaker plays an error warning sound (Siren Vintage Sounds). In the experiment, the MCU continuously raises questions. This fatigue-relieving operation mode (MRM) was repeatedly executed until the end of the driving experiment in the MRM. In the experiment, the data acquisition needed to stop responding to eliminate interference.

Additionally, before the start of the experiment, we needed to judge whether the extra increase of the cognitive load (arithmetic operation) induced by the MRM will affect the subjects’ ability to cope with unexpected problems. For 12 subjects, each of them needed to drive for 1 h in the two driving modes (the MRM and normal driving mode), respectively. When the driving time reaches 1 h in one driving mode, the subjects need to do an emergency test on the emergency stop of the vehicle in front of them and record the reaction time. The emergency test in the MRM can only be carried out after all the subjects have completed the test in the normal driving mode. The test period was from 13:00 p.m. to 16:00 p.m.

### 2.3. Methods

In the experiment, EEG signals were collected every half an hour, as shown in Figure 2A. Then, the EEG signals were preprocessed by wavelet packet decomposition (WPD), as shown in Figure 2B. Finally, the preprocessed EEG signals were used to analyze the characteristics of the brain network, eye movement, and power spectrum, respectively. The whole process of EEG signal processing in this paper is shown in Figure 2.

#### 2.3.1. Statistical Analysis Algorithm

The statistical analysis method (two-tailed *t*-tests) was applied in our study. Using this method, we compared the differences in detection results. In the comparative analysis section, the two-tailed test was used to identify significant differences in driving fatigue between the two driving modes (the MRM and the normal driving mode).

#### 2.3.2. Signal Preprocessing

The EEG signal is a weak electrical signal, which is extremely vulnerable to external interference. Therefore, denoising is needed before signal analysis. In our experiment, we chose the WPD, which can provide more sophisticated analysis of signals, to deal with the noise. Additionally, the WPD can also select a suitable frequency band to match the signal spectrum according to the characteristics of signal analysis, which can reflect the essential characteristics of the signal. The formula of WPD is shown as follows:

Assume that the original signal is expressed as *f*(*t*). We obtained 2*i* sub-bands in the *i* points class after WPD. The source signal *f*(*t*) can be expressed as:(1)$$f(t)=\sum_{j=0}^{2^{i}-1}f_{i,j}(t_{j})=f_{i,0}(t_{0})+f_{i,1}(t_{1})+\cdot \cdot \cdot +f_{i,2^{i}-1}(t_{2^{i}-1})$$
in which, *j* = 0, 1, 2, …, 2*^i^* − 1, *f_i_*_,*j*_(*t_j_*) is the reconstruction of signals in the ith layer node (*i,j*) when using wavelet packet decomposition. According to Parseval theorem and Equation (1), the energy spectrum of the signal *f*(*t*) after WPD can be calculated and obtained:(2)$$E_{i,j}(t_{j})=\int_{\Gamma }^{}|f_{i,j}(t_{j}){|}^{2}dt=\sum_{k=1}^{m}|x_{j,k}{|}^{2}$$
in which, *E_i_*_,*j*_(*t_j_*) is the band energy by which *f(t)* was decomposed to node (*i,j*) using WPD. *x_j_*_,*k*_ (*j* = 0,1,2,…,2*^i^*−1; *k* = 1,2,…,*m*) is the discrete points amplitude of the reconstructed signal *f_i,j_*(*t_j_*). *m* is the signal sampling points. In our study, we conducted 4-layer decomposition of the band to extract θ (4–8 Hz), α (8–14 Hz) and β (14–32 Hz) rhythms.

#### 2.3.3. Correlation Coefficient

In statistics, the Pearson correlation coefficient (Pearson product-moment correlation coefficient, abbreviated as PPMCC or PCCs) is a method for measuring the relationship between two variables. In this paper, we use this method to analyze the relationship between any two channels of EEG signals. The Pearson correlation coefficient is defined by:(3)$$r_{XY}=\frac{E[(X-E(X))(Y-E(Y))]}{\sigma_{X}\sigma_{Y}}=\frac{E(XY)-E(X)E(Y)}{\sigma_{X}\sigma_{Y}}$$
in which, *E*(·) is the expected value operator, and σ*_X_*, σ*_Y_* are the standard deviation (SD). In the present situation, we analyzed series consisting of n samples of data. Accordingly, the correlation coefficient was computed by:(4)$$r_{XY}=\frac{1}{n-1}\sum_{i=1}^{n}\left(\frac{x_{i}-\bar{x}}{\sigma_{X}})(\frac{y_{i}-\bar{y}}{\sigma_{Y}}\right)=\frac{\sum_{i=1}^{n}x_{i}y_{i}-n\bar{x}\bar{y}}{(n-1)\sigma_{X}\sigma_{Y}}$$
in which, $\bar{x}$ and $\bar{y}$ are the series means, and σ*_X_*, σ*_Y_* are the standard deviation.

#### 2.3.4. The Complex Brain Networks

Previous studies have shown that the method of analyzing brain network connectivity has been proven to be a very effective and informative way to analyze brain function and mental state [28,29,30]. In this study, we used the two parameters of the complex network, clustering coefficient and global variables, to analyze the brain functional differences of subjects. These two parameters are described in the following sections.

Clustering Coefficient

For node *i* of a complex network, which is expressed as the ratio of the number of existing edges to the number of maximum possible edges between the neighbors of *i* [31,32], the clustering coefficient can be represented as:(5)$$C_{i}=\frac{E_{i}}{D_{i}(D_{i}-1)/2}$$
in which, *E_i_* is the number of existing edges between the neighbors of node *i* and *D_i_* is the degree of connectivity of node *i*.

Global efficiency

The higher the value of the complex network’s parameter G, the faster the information transmission is. L*_i,j_* is the path length between two nodes *i* and *j*, which is the minimum number of edges needed to connect. L*_i,j_* is mathematically defined as Equation (6) [31,32]:(6)$$L=\frac{1}{N(N-1)}\sum_{i,j=1,i\neq j}^{N}D_{ij}$$
in which, *D_ij_* is the minimum path length *L* between nodes *i* and *j*. *N* is the number of nodes within a network. The global efficiency of nodes can be defined by:(7)$$G=E_{global}=\frac{1}{N(N-1)}\sum_{i\neq j\in G}\frac{1}{L_{i,j}}$$
in which, *L_i,j_* is the minimum path length between nodes *i* and *j*. *N* is the number of nodes within a network. From Equation (7), it can be concluded that a network, which is characterized by a short minimum path length between any pair of regional nodes, has high global efficiency [33,34].

In this paper, the correlation coefficients between pairs of signals from 14 channels were calculated using Equation (4). Then, brain networks were constructed according to the following steps.

The sub-band signals θ (4–8 Hz), which were extracted from the denoised EEG signals (Figure 3A), were used to construct the adjacent matrix (Figure 3B). Then, a reasonable threshold value T was determined, which was used to determine the edge connection between two nodes. This means that there was an edge connection between node *i* and node *j* if the correlation coefficient between them was greater than the threshold value T, otherwise, no edge existed between nodes *i* and *j*. Finally, the networks were formed (Figure 3C).

#### 2.3.5. The Relative Power Spectrum

The four sub-band signals of human EEG signals δ (0–4 Hz), θ (4–8 Hz), α (8–13 Hz), and β (13–35 Hz) are particularly useful for characterizing brain activity. Their different power spectrum ratios, θ/β, θ/α + β, θ + α/β, θ + α/α + β, and β/α, are often used to analyze people’s mental fatigue characteristics, which can also show different characteristics of driving fatigue [31,35,36]. In this paper, we used the ratio β/(θ + α) to analyze driving fatigue.

#### 2.3.6. Subjective Questionnaire

Previous studies have shown that the subjective questionnaire (SQ) is an effective method to evaluate people’s mental fatigue [37,38]. In this paper, we used the 7-point Samne Perelli Fatigue Scale (1-Fully alert, wide awake, 2-Very lively, responsive, but not at peak, 3-Okay, somewhat fresh, 4-A little tired, less than fresh, 5-Moderately tired, let down, 6-Extremely tired, very difficult to concentrate, 7-Completely exhausted, unable to function effectively) to evaluate the driving fatigue of the subjects. At each stage of the experiment, subjects were asked to rate their subjective fatigue and give a score.

## 3. Results

### 3.1. Subjective Questionnaire

The subjective questionnaire is a common subjective way to evaluate human mental fatigue [38,39]. Figure 4 shows the variation tendency of the average questionnaire scores for the 12 subjects at the 7 driving stages.

As can be seen from Figure 4, the average SQ score of subjects driving in the two test modes presented an increasing trend, which means that the subjective driving fatigue degree of subjects continued to deepen over time. In addition, there was no significant difference in the average SQ score between the two driving modes from driving stage 1 to stage 4. However, the subjective fatigue degree of the subjects in the last three driving stages (stages 5–7) was obviously lower in the MRM than in the normal driving mode, which means that the MRM can alleviate driving fatigue compared with normal driving.

### 3.2. The Response Error Rate of Subjects

In our experiment, with the prolongation of driving time, the subjects experienced different degrees of driving fatigue and their alertness also decreased. When the response time of a subject exceeded 5 s, the system thought that the response was overtime, and when the subjects answered incorrectly, the system recorded the answer as wrong. Both cases were recorded by MCU within 27 min of each experimental stage. The results are shown in Figure 5 below.

Figure 5 shows the response error rates of subjects in this experiment. From the data in the figure, it can be concluded that with the driving time, the reaction time of the subjects increased, and some of them had already exceeded the time limit. In addition, the response error rate also showed an upward trend over time. This means that although this driving mode can combat driving fatigue, driving fatigue will still become more and more serious with the extension of driving time.

### 3.3. Brain Network Analysis

#### 3.3.1. Choice Threshold (T)

Previous studies have shown that when people gradually enter a state of mental fatigue, the EEG activity characteristics in the human brain will change accordingly [39]. Craig et al. found that the activities of θ and α increased with the gradual increase of human mental fatigue [32]. Belyavin and Wright observed an increase in the θ activities and a decrease in the β activities when a human gradually became mentally fatigued and their alertness decreased [40]. Other studies have also shown an increase in θ activity during mental fatigue [36,41,42]. Summing up the research results of the above studies, we can conclude that when humans gradually become mentally fatigued, their alertness decreases along with an increase in θ activity (see Section 4.2 for details). Therefore, we used the θ sub-band signals to analyze driving fatigue using the brain network.

In order to analyze the differences in brain network features constructed using different thresholds, networks were formed at all the thresholds for each stage. In our study, we conducted a comparison of the whole range of values of T, 0.01 < T < 0.51, with increments of 0.025, and repeated the calculation for each value of T. In general, the selection of the threshold value should depend on the research problem and should fall within the scope of an educated guess [43]. Figure 6 shows the comparison of the coefficient C averaged over subjects in the driving stage 1 with that of other stages (driving stages 2–7).

Figure 6 shows the changes in the clustering coefficient C for different brain networks. Obviously, the changes in C showed a downward trend in all driving stages. The contrast difference was significant when the threshold value was chosen in the numerical interval 0.235~0.46 (stage 1–2:|*t*| = 4.151 > *t*_0.05,18_ = 2.101, *p* = 0.001 < 0.05; stage 1–3: |*t*| = 3.653 > *t*_0.05,18_ = 2.101, *p* = 0.002 < 0.05; stage 1–4: |*t*| = 3.492 > *t*_0.05,18_ = 2.101, *p* = 0.003 < 0.05; stage 1–5: |*t*| = 3.983 > *t*_0.05,18_ = 2.101, *p* = 0.001 < 0.05; stage 1–6: |*t*| = 4.091 > *t*_0.05,18_ = 2.101, *p* = 0.001 < 0.05; stage 1–7: |*t*| = 4.772 > *t*_0.05,18_ = 2.101, *p* = 0.001 < 0.05).With this method, the threshold interval (0.26~0.435) with significant differences in global efficiency was determined. We can conclude that the parameter (G, C) difference in the brain network is obvious when the threshold interval is selected to be 0.26~0.435. In our study, the mean value of the threshold interval 0.26~0.435 was calculated to be T = 0.3475. The brain network corresponding to each stage of the driving experiment was constructed. The results are shown in Figure 7.

#### 3.3.2. Network Characteristics

In this paper, we built brain networks for subjects driving in normal mode and the MRM according to the steps in Figure 3. Figure 7 shows the brain network for the subjects at 7 stages (stages 1–7).

Figure 7 shows the change trend of the brain network for subjects following extended driving time under two experimental driving modes. There were no significant differences in the brain network between the two driving modes in the first three driving stages. However, after driving stage 3, there was a significant difference in the brain network between the two driving modes (*t* = 2.719 > *t*_0.05,6_ = 2.447, *p* = 0.04 < 0.05). The brain network parameters C and G were calculated by Equations (5) and (7), respectively. The change tendencies of the mean values of the two parameters are shown in Figure 8.

From Figure 8, we can clearly see that these two brain network parameters (C and G), which relate to the brain networks of the subjects in two driving modes (driving in normal mode and in the MRM), showed increasing trends over time. Moreover, these two brain network parameters (C and G) showed significant differences in the two driving experimental modes (C: *t* = 2.678 > *t*_0.05,12_ = 2.179, *p* = 0.02 < 0.05; G: *t* = 2.289 > *t*_0.05,12_ = 2.179, *p* = 0.04 < 0.05). The growth trends of the parameters (C and G) of subjects driving in MRM were slower compared with subjects driving in normal driving mode.

### 3.4. The Relative Power Spectrum Ratio

Previous studies have shown that it is very common to analyze people’s mental fatigue state by using the relative power spectrum of EEG signals. The variation tendency of the relative power spectrum ratio β/(θ + α) over time is shown in Figure 9.

Figure 9 shows that the ratio (β/θ + α) of the relative power spectrum presented a downward tendency over time, which indicates that the level of brain activity decreased over time. The driving fatigue degree of subjects gradually decreased over time. Additionally, the ratio β/θ + α, which relates to the MRM, showed a smaller downward trend compared with driving in the normal driving mode. The difference between the two driving modes is significant (P3: |*t*| = 2.527 > *t*_0.05,12_ = 2.179, *p* = 0.03 < 0.05; P4: |*t*| = 2.633 > *t*_0.05,12_ = 2.179, *p* = 0.02 < 0.05). This means that the driving fatigue of subjects in the MRM decreased more slowly compared with that of the normal mode.

### 3.5. Eye Movement

Studies have shown that humans make coordinated movement of the eyes to orient the high-acuity part of the retina on relevant objects in the world and facilitate their processing [12,13,44]. Human eye movement signals are easily detected in the prefrontal region of the brain. In this paper, channels F3 and F4, in which the signal waveforms have a negative correlation when eye movements occur, were chosen to identify eye movement features. Taking eye movement to the right as an example, the waveforms of the two leads F3 and F4 are shown in Figure 10.

In the experiment, a moving window with a width of 20 samples was established, and the eye movement signals were identified using Equation (8):(8)$$K_{i}=\frac{y(x_{i+20})-y(x_{i})}{20}$$

The absolute value of K, which reflects the signal fluctuation characteristics of F3 and F4, is greater than 2 when eye movement occurs. At the same time, the signal waveforms of F3 and F4 show a negative correlation when eye movement occurs, and the absolute value of the correlation coefficient is greater than 0.85. In our work, K and the correlation coefficient *r* were used to identify eye movements. The discrimination logic is shown in Figure 11.

The logical relationship for eye movement recognition is illustrated in Figure 11. The recognition results of eye movement are shown in Figure 12.

Obviously, Figure 12 shows that the number of eye movements per minute presented an overall downward trend, which means that the two driving modes (driving in MRM and driving in normal mode) can gradually deepen the driving fatigue of subjects. In addition, the number of eye movements, which relates to the MRM, showed a smaller downward trend compared with driving in normal driving mode and the difference between the two driving modes was significant (*t* = 2.288 > *t*_0.05,12_ = 2.179, *p* = 0.04 < 0.05). This indicates that the driving fatigue of subjects in MRM decreased more slowly compared with in the normal mode.

## 4. Discussion

Previous studies have shown that people in a state of driving fatigue show abnormal external physiological characteristics such as unresponsive behaviors, decreased vigilance and errors in judgment, which seriously threatens traffic safety [45,46,47]. Traffic accidents caused by driving fatigue account for a considerable proportion of all accidents [48,49]. Hence, it is necessary to effectively relieve driving fatigue. Studies have shown that repetitive and monotonous external environmental information can easily lead human beings to be in a state of mental fatigue, in which our brain activity is inhibited [50,51,52]. In our study, we tried to stimulate human brain nerves repeatedly to keep them active all the time to combat mental fatigue.

### 4.1. Brain Network

Previous studies have shown that the characteristics of the brain network change with the changes of human alertness [53,54,55]. Kar and Sun indicated that there is a growth trend of the two parameters (C and G) as subjects’ fatigue deepens [33,56]. Stam et al.’s study showed that human mental fatigue continues to deepen during the continuous operation processes, and the corresponding brain network parameters C and G show overall upward trends [57]. This result is consistent with our research. In this study, as shown in Figure 8, the two brain network parameters (C and G) of the subjects driving in normal mode also showed overall upward trends. However, the subjects driving with MRM showed slower upward trends because the brain nerves were continuously stimulated at successive driving stages, which keep the nerves in a relatively excited state.

### 4.2. The Relative Power Spectrum Ratio

Previous studies have shown that when people gradually enter a state of mental fatigue, the EEG activity characteristics in the human brain will change accordingly [39]. Craig et al. found that the activities of θ and α increased with a gradual increase in human mental fatigue [32]. Belyavin and Wright observed an increase in θ activities and a decrease in β activities when humans gradually became mentally fatigued and their alertness decreased [40]. According to the study by Torsvall and Åkerstedt, an increase in α activities is the most sensitive indicator of mental fatigue [58]. Other studies have also shown an increase in θ activity during mental fatigue [36,41,42]. Summing up the research results of the above researchers, we conclude that when humans gradually become mentally fatigued, their alertness decreases along with increases in θ and α activities and a decrease in β activities. Based on this viewpoint, the ratio (β/θ + α) will decrease with the deepening of mental fatigue, which is consistent with our research results, as shown in Figure 9.

### 4.3. Eye Movement

In our experiment, the eye movement signals were very obvious. When people perform eye movements (move to right or move to left), the voltage changes in opposite directions will occur in symmetrical brain regions and corresponding EEG waves will also show fluctuations in opposite directions. In our study, for eye movement signals, the timestamp from the beginning of the fluctuations to the stable fluctuations was at least 20, as shown in Figure 10. At this time, the K value is calculated by Equation (8), whose absolute value was at least 2. In our study, we chose a time window of 20 and |K| > 2 to detect eye movements. The eye movement signals, whose waves showed opposite characteristics in symmetrical brain regions, were identified using the Pearson correlation coefficient. In this paper, taking the time width as 20, the fluctuation characteristics of the eye movement signals were calculated, and the absolute minimum value of the Pearson correlation coefficient for two symmetric channel (F3 and F4) signals was 0.85. Therefore, we chose this condition (|K| > 2 and |*r*| > 0.85) to judge eye movement.

Studies have shown that eye movement frequency is related to mental fatigue [59,60]. Marzano et al. studied human eye movements and concluded that when humans gradually enter a state of mental fatigue, their eye movements become significantly slower [61]. The research organized by Shin et al. showed that when subjects’ driving fatigue gradually deepened, their eye movement frequency decreased obviously [62]. Cazzoli et al. and Russo et al. concluded that people’s ability to obtain external information from the eyes was weakened when their mental fatigue continued to deepen, which was characterized by a decrease in eye movement frequency [63,64]. These research conclusions are consistent with our experimental results. In this study, as shown in Figure 12, the eye movement frequency of subjects driving in normal mode also showed an overall downward trend. However, the eye movement frequency of subjects driving with MRM showed a relatively slow downward trend. The reason for the difference was that the MRM stimulated the brain nerves, causing subjects to maintain their normal thinking activities, which meant that the brain nerves maintained their ability to obtain outside information.

From the above comparative analysis, we can conclude that human bioelectric signals (EEG and EOG) can effectively be used to monitor human driving fatigue and the MRM proposed in this paper can effectively relieve driving fatigue caused by long-term monotonous driving. Thus, it can play an important role in alleviating the symptoms of mental fatigue.

### 4.4. Previous Studies and This Study

Many traditional methods can be used to relieve human mental fatigue, such as stimulating acupuncture points, reducing the intensity of work and having a short rest. Although using electricity to stimulate acupuncture points of the human body can effectively combat mental fatigue, electrical stimulation has some unavoidable side effects on the human body. Other traditional methods of relieving fatigue usually require parking, which may not be feasible for long-distance driving. It is uncertain whether there are side effects of relieving mental fatigue by long-term electrical stimulation of human acupoints. Oron-Gilad et al. demonstrated that completing an appropriate number of cognitive tasks, especially ones that involved understanding and memory, could keep drivers alert [21]. Verwey and Zaidel indicated that performing an attention-demanding secondary task can effectively improve alertness when they drove for a long time [22]. Drory’s experimental result showed that a non-mandatory driving-related task of a reasonable size could make a driver maintain a high level of alertness [23]. An appropriate increase in cognitive activity by factors other than driving can indeed effectively combat driving fatigue [21,22,23]. In this paper, we used this method to relieve driving fatigue. However, we used mental arithmetic combined with reward and punishment mechanisms to stimulate human brain nerves in order to effectively relieve mental fatigue. The equipment used in our experiment is simple, portable and inexpensive. Normal driving is not affected when the equipment is used to resist fatigue, and no sides effects are caused. The MRM method of resisting driver fatigue is carried out online and in real time, which is of great significance for future practical applications.

In our experiment, we found that human eye movement signals were easily detected in the prefrontal region of the human brain, and in symmetrical areas of the human brain, the wave signals of eye movement were in opposite directions. Although eye movement signals can also be recognized by selecting these two electrodes (one vertical and one horizontal), the signals of these two channels fluctuate unidirectionally in the time domain. The simple method Equation (8) used in our manuscript cannot distinguish unidirectional fluctuations from common interference signals, such as blinking. To measure the signals in the prefrontal region of the human brain in this paper, we chose two symmetrical channels (F3 and F4), for which the signal waveforms have a negative correlation when eye movements occur. We used the Pearson correlation coefficient algorithm to identify eye movements.

In addition, the driving fatigue process is a process of physiological function change, and its change is not completed instantaneously [65]. Previous experiments show that the driving fatigue test generally lasts for 3–5 h, and the fatigue change curve also changes regularly with time [32]. The research by Rongrong demonstrated that the fatigue parameter (posterior probabilities) showed an overall upward trend over time [3]. In the study of Luo, 5′ EEG signals, which were used to analyze the variation characteristics of driving fatigue, were saved in each experimental period [52]. In the whole experimental process, the experimental stages are evenly divided, and in each experimental stage, the subjects’ driving signals are detected for a short time. Then, the change trend of driving fatigue in the whole experimental stage can be effectively detected [32]. Based on the above experimental research methods, we chose the method of collecting three-minute EEG signals every half hour to carry out this research.

### 4.5. Limitations

In this study, we used the MRM to relieve driving fatigue caused by long-term monotonous driving. And we also studied whether the extra increase of the cognitive load (arithmetic operation) induced by the MRM will affect the subjects’ ability to cope with unexpected problems. The results show that the average time taken by the subjects to respond to the emergency stop of the vehicles ahead is 0.53 s in the MRM and 0.87 s in normal driving mode. Moreover, the standard deviation of reaction time is 0.09 s in the MRM and 0.16 s in normal driving mode. There was a significant difference in the emergency response time between the two driving modes (|*t*| = 6.108 > *t*_0.05,22_ = 2.074, *p* = 0.00 < 0.05). It means that the extra increase of the cognitive load in this study will not affect the subjects’ ability to cope with unexpected problems. Although this method was proven to be effective in relieving long-term driving fatigue, only a simple mathematical operation was used in our work to stimulate cranial nerves to make them active. Whether more complicated arithmetic operations are more conducive to relieving or aggravating mental fatigue has not yet been studied.

Additionally, the selected threshold T in this paper is a common value for all subjects (10 males and 2 females; aged 32 ± 1.6 (SD)). The T value in Figure 4 is a grand average between the subjects. In the future, when the brain network method is applied to detect the driver’s driving fatigue state, the selected threshold value T (T = 0.3475) will change due to the differences of individual drivers or driving environment and other factors. Therefore, the brain network threshold T determined in this paper is not applicable to all users.

### 4.6. Future Research Lines

Driving drowsiness is a state of physical fatigue that describes serious driving fatigue caused by continuous long-term driving. At this time, the driver must stop driving for a rest. In future research, our work will be mainly divided into three aspects. One is choosing different algebraic operations to determine the relationship between the amount of cognition required other than driving and the effect of relieving mental fatigue. The next is comparing the MRM method with existing effective fatigue relieving methods, such as electrode shifts for head-resting, to further improve these methods. The last one is using our method to alleviate driving fatigue in real driving.

## 5. Conclusions

In the present work, the MRM method was proposed for alleviating driving fatigue. Characteristics of the complex brain network, which can effectively reflect brain activity information, were used to detect the change of driving fatigue over time. Combined with the traditional eye movement characteristics and the SQ, the changes of driving fatigue characteristics were comprehensively analyzed. The results show that driving fatigue can be effectively relieved using the MRM, which means that rhythmic stimulation of brain nerves for thinking activities can slow down the rate of fatigue development. In addition, the response equipment is low cost and practical; therefore, it has the potential to be used in actual driving situations in the future.

## Figures and Tables

**Figure 1 sensors-19-04883-f001:**
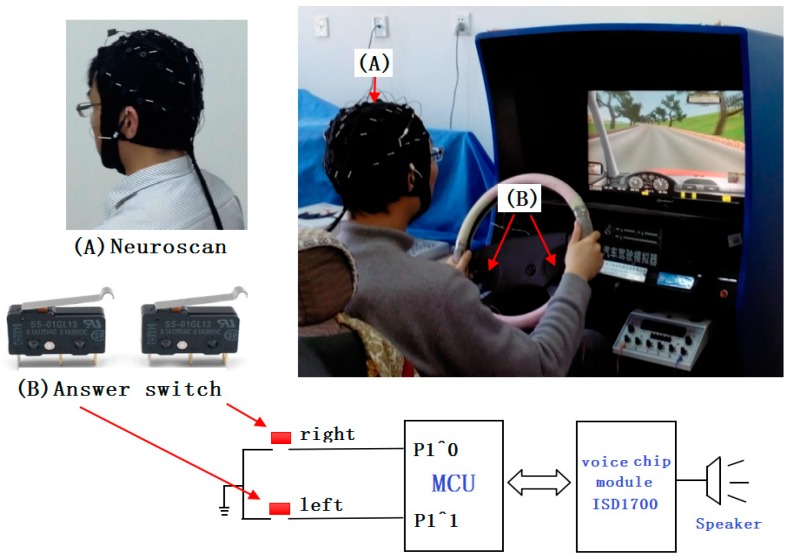
The experimental set-up.

**Figure 2 sensors-19-04883-f002:**
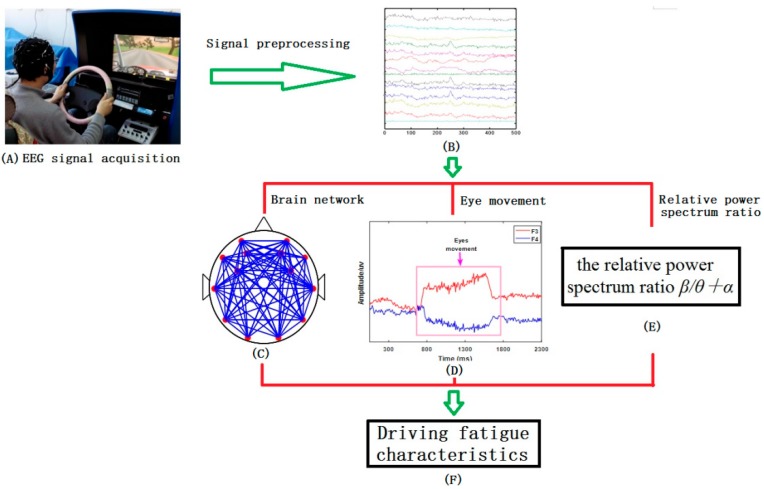
(**A**) EEG signal acquisition; (**B**) 14 channels of original signals collected; (**C**) Building brain network; (**D**) Analysis of eye movement signal characteristics; (**E**) Calculate the relative power spectrum ratio β/θ+α; (**F**) Analysis of Driving Fatigue Characteristics.

**Figure 3 sensors-19-04883-f003:**
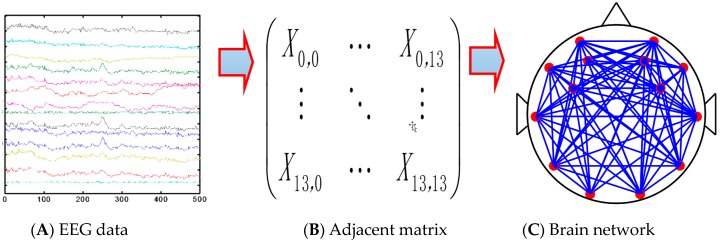
Steps of the construction of brain network.

**Figure 4 sensors-19-04883-f004:**
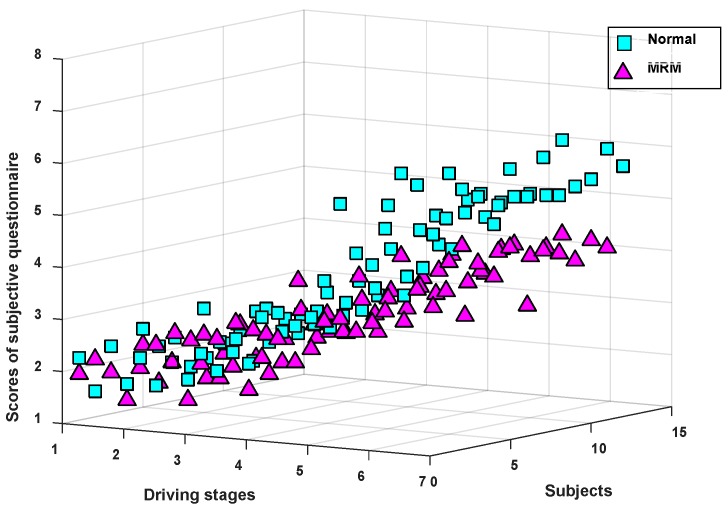
Subjective questionnaire (SQ) scores (mean ± standard deviation (SD)) for the 7 driving stages.

**Figure 5 sensors-19-04883-f005:**
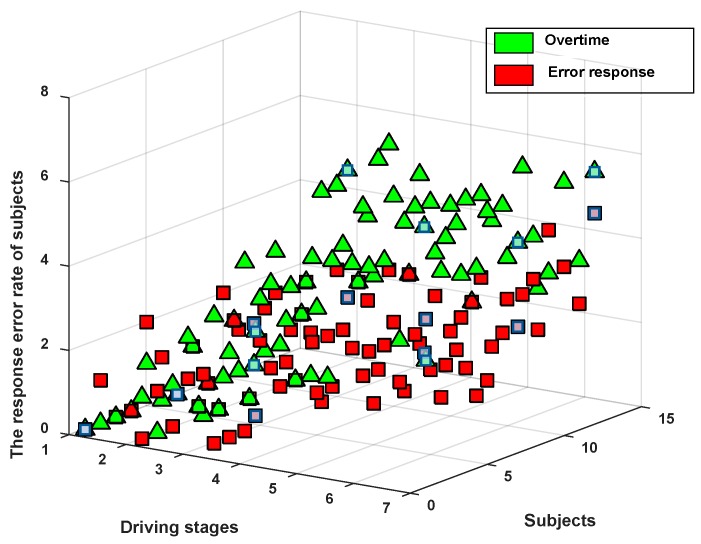
The response error rates of the subjects.

**Figure 6 sensors-19-04883-f006:**
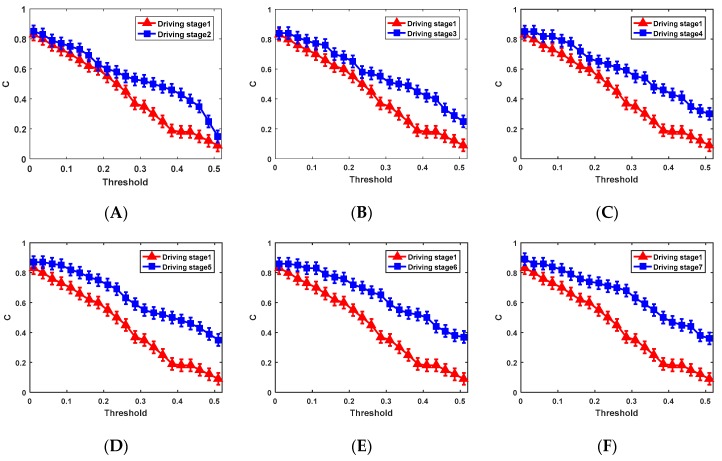
(**A**) Comparison of the C between driving stage 1 and driving stage 2 when the threshold value take different values; (**B**) comparison of the C between driving stage 1 and driving stage 3 when the threshold value take different values; (**C**) comparison of the C between driving stage 1 and driving stage 4 when the threshold value take different values; (**D**) comparison of the C between driving stage 1 and driving stage 5 when the threshold value take different values; (**E**) comparison of the C between driving stage 1 and driving stage 6 when the threshold value take different values; (**F**) comparison of the C between driving stage 1 and driving stage 7 when the threshold value take different values.

**Figure 7 sensors-19-04883-f007:**
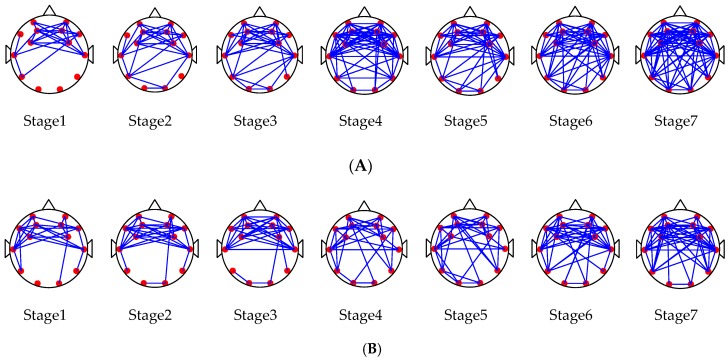
(**A**) Changes of Brain Network Connections at all driving stages (1–7) in normal driving mode; (**B**) Changes of Brain Network Connections at all driving stages (1–7) in Man-machine Response mode.

**Figure 8 sensors-19-04883-f008:**
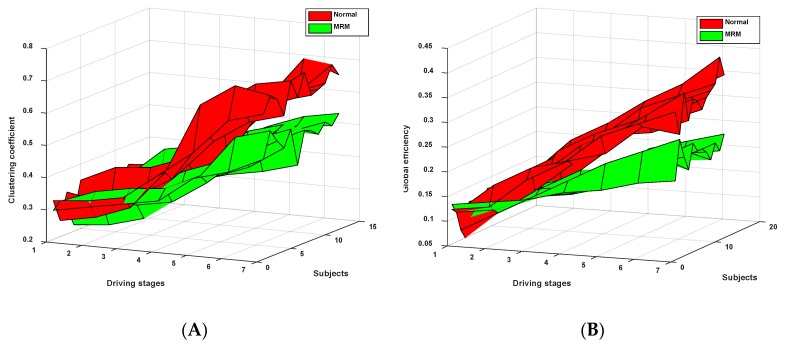
(**A**) Variation tendency of the cluster coefficient C at 7 driving stages; (**B**)Variation tendency of the global efficiency G at 7 driving stages.

**Figure 9 sensors-19-04883-f009:**
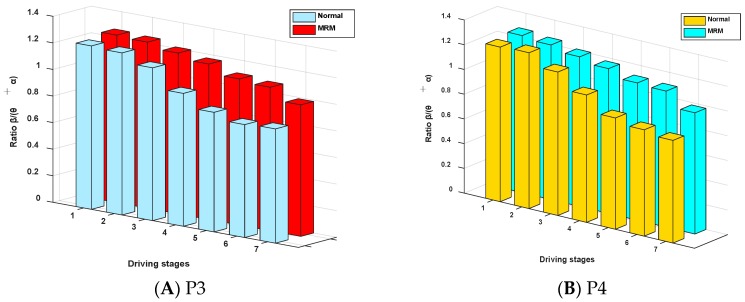
The ratio β/θ + α of the relative power spectrum.

**Figure 10 sensors-19-04883-f010:**
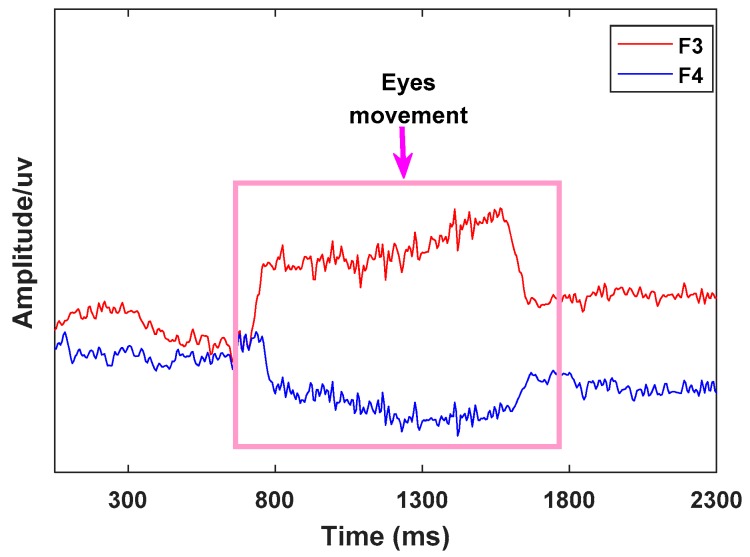
Eye movement signals.

**Figure 11 sensors-19-04883-f011:**
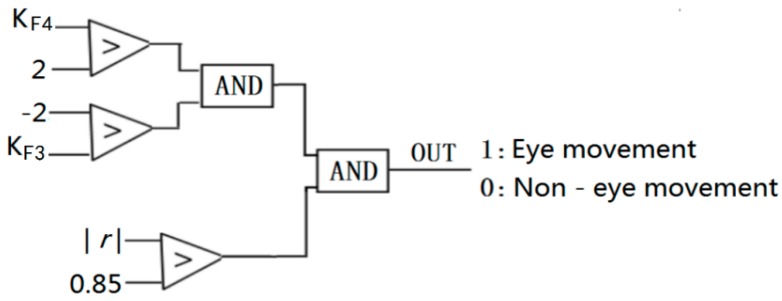
The logic of motion recognition.

**Figure 12 sensors-19-04883-f012:**
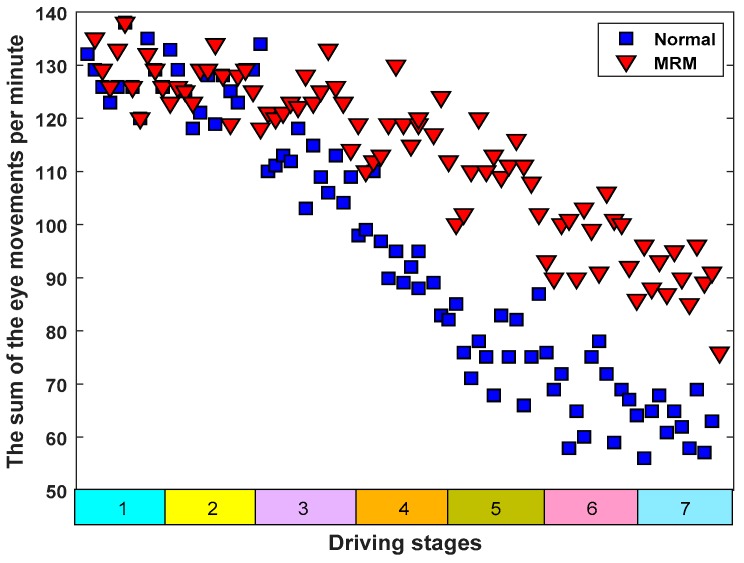
The sum of the eye movements of each driver in the experiments.

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
