# Peer review of "Study on the Effect of Man-Machine Response Mode to Relieve Driving Fatigue Based on EEG and EOG"

_sensors, 2019, doi:10.3390/s19224883_

Round 1
Reviewer 1 Report
This manuscript presents a set up to monitor fatigue during driving recordings from an electroencephalogram (EEG) recording to assess three driving fatigue characteristics (EEG signal complex network parameters, eye movement and sub-band signals from the EEG's power spectrum). Interestingly, a subjective questionnaire was applied to assess mental fatigue the level of fatigue (or stage) on a scale from 1 to 7 (from fully alert to completely exhausted). The authors tested the monitoring procedure during two conditions in each of 12 participants: a man-machine response mode (MRM) where de driver interacted with a micro-control unit (MCU) and had to provide answers to the MCU through an answer switch, and a normal mode where there were no interactions with the MCU. The results show less fatigue during the tests that included interaction with the MCU during the MRM driving compared to the normal driving mode. This approach is interesting and original, and the manuscript is well written.
Author Response
Dear reviewer,
Thank you for your comments and suggestions for our manuscript.
Revision01
This manuscript presents a set up to monitor fatigue during driving recordings from an electroencephalogram (EEG) recording to assess three driving fatigue characteristics (EEG signal complex network parameters, eye movement and sub-band signals from the EEG's power spectrum). Interestingly, a subjective questionnaire was applied to assess mental fatigue the level of fatigue (or stage) on a scale from 1 to 7 (from fully alert to completely exhausted). The authors tested the monitoring procedure during two conditions in each of 12 participants: a man-machine response mode (MRM) where de driver interacted with a micro-control unit (MCU) and had to provide answers to the MCU through an answer switch, and a normal mode where there were no interactions with the MCU. The results show less fatigue during the tests that included interaction with the MCU during the MRM driving compared to the normal driving mode. This approach is interesting and original, and the manuscript is well written.
Thank you for your careful review for our manuscript.
Reviewer 2 Report
In this paper the authors report a study on driving drowsiness. Specifically, they test whether a “Man-machine Response Mod” (MRM) system, which stimulates the driver by posing arithmetical problems, can alleviate driving drowsiness. The effects are assessed by a number of different measures (including EEG recordings, self-report and eye movements). All these measures seem to agree in showing a decreased drowsiness when driving was done with the MRM system, compared to a control condition. While the experiment is potentially interesting, some important details are unclear or not reported, which makes it difficult for the reader to assess the soundness of the results and their implications (see below). For example, the reporting of statistical tests is incomplete. Additionally, I think the MRM system proposed and the experimental protocol used may have some shortcomings, which are not discussed in the current version of the article. Finally, the text contains some awkward wording and sentences, and could be improved for clarity (some examples and suggestions below). Concerning the shortcoming of the MRM, while I am convinced that continuously presenting subjects with arithmetical problems, music and other auditory stimulation might increase vigilance and prevent to some extent drowsiness, I am not convinced it is a good idea to do it while driving in real situations. I suspect that the cognitive load induced by these tasks might make the driver less attentive to the road and, for example, slow down its reaction times in case of danger (as it’s been showed by psychological research on dual-tasking). The authors state in the discussion that “Normal driving is not affected when the equipment is used to resist fatigue, and no sides effects are caused.” however this statement cannot be supported by the data, since there are no measures of driving performance. This point could be addressed, for example, by adding in the simulation some “danger” situations that needs a prompt reaction by the driver (e.g. an infraction by another car). If the reaction times to these unexpected events did not differ between MRM mode and normal driving, then the authors could suggests that the MRM mode did not interfere with driving, but at present this statement seems unwarranted. This also present a significant limit for the application of the proposed technology in real driving, however it is not discussed. Finally, I think that the experimental protocol might be confounded. In the MRM condition participants were continually stimulated with acoustical stimuli (numbers, cheerful music, error warnings). This aspect of the task is not matched in the control condition. Is the reduced drowsiness due to the executing the arithmetic tasks, or simply due to the auditory stimulation? Perhaps simply listening to music while driving would reduce drowsiness as much as the MRM mode (or perhaps even more?). Because the auditory stimulation was not matched, this question cannot be addressed from the current results. Specific comments: - Some aspects of the experimental protocol are unclear or not reported. For example: I assume that each participant run the MRM and normal on different days, and at page 2, lines 66-67, it is written that the experiment was divided in two “types”: My understanding was that subjects were divided in two groups, and the order of the two condition (MRM and control) counterbalanced. Yes this is not very clear and should be stated more clearly. In the next page it is stated that half-hour of sleep was arranged (pag 3 lines 92), however the time indicated within parenthesis (0:00 – 0:30) seems wrong and do not allow to collocate the sleep break in the experiment (which is reported to start at 13pm and ends at 16pm for each participants). I found that the description of the MRM system itself (page 4 lines 117-126) lacks details. If I understood correctly, the participant were asked to whether the sum of two numbers exceeded 100 or not, and then say whether the sum was divisible by 3. How were the digits generated? What was their range of possible values? Did the system formulated an explicit question, or simply presented the two numbers? Were the participants asked to complete both responses within 5 seconds, or were they allowed 5 second for each response? Did the system continuously proposed questions, without any breaks, or was it interleaved with pauses? What sounds were used as cheerful music and error warning signal? - Figure 4: in the text is written that the figure represents the “contrast” C (actually the clustering coefficient) at the two stages for each subject? However the figure seems to show the coefficient C averaged over subjects in the stage 1 of driving compared to the other stages (e.g. panel 1 is stages 1 vs 2, panels 2 is stage 1 vs 3, etc.). There is an inconsistency between what is written in the text and the figure. - Reporting of statistics: please report the statistic (e.g. t of F values) together with the p-value. Also, report the actual value of the p-value, not simply whether is exceed or not the conventional alpha threshold. - It is not clear exactly what test has been used to assess the differences in network characteristics (page 8). For example at lines 257-258 one unique test is reported to indicate that “obviously” there was no difference in the brain network in the first three stages (please remove the word “obviously” from the text; if it was obvious then there should be no need to assess it with a statistical test). Moreover, it is reported that “p<0.05” even though the text indicate that there was no difference, so there is an inconsistency between the p-value and the text. - power spectrum ratio (and figure 7). Why the figure only shows data from the parietal electrode P3 and P4? Was there a difference only in these electrodes? Was the difference significant? (No statistical test is reported here). - analysis of eye movement. It is difficult to assess the formula used (formula 8) because the sampling rate of the signal is not reported, so it is not clear to what temporal interval the 20 samples correspond to. - The text is in several instance redundant (e.g. pag 2, lines 71-74 with pag. 3 lines 104-109), and also the paragraph at page 14 (from lines 406) is almost identical to a paragraph in the introduction (page 2 from line 43), with the exception that the reference numbers are now wrong (e.g. Zhao et al, at line 410, is reference number 19, not 20). - line 452: “division”, actually is both addition and division I should disclose that I am not a native speaker, however I couldn’t help but notice that the text contain some awkward phrasing and could be improved for clarity in several places. Here are some suggestions: - “standard values of deviation” (page 5) is usually referred to simply as “standard deviation” - pag 7, lines 202: “contain the main information about the brain” sounds a bit awkward and should perhaps be replaced with something like “are particularly useful for characterizing brain activity” - pag 7, line 224, “above researchers” should be “above studies” - pag 7, line 231, “well-founded speculation” seems an oxymoron and should perhaps be replaced with “educated guess” or something similar - line 280, “brain nerve activity” should just be “brain activity” - pag 10, lines 188, “Studies have shown that the two eyeballs move synchronously to obtain information when human beings recognize information form the outside world” is awkward phrasing. Something like “Humans make coordinated movement of the eyes to orient the high-acuity part of the retinal on relevant objects in the world and facilitate their processing” or something along these lines might be better. - line 327, maybe “prevent” should replace “resist” here - line 404, “inhibiting the deepening speed of mental fatigue” also sounds awkward. Maybe could be replaced with “alleviate the symptoms of mental fatigue” or “prevent the accumulation of mental fatigue”
Author Response
Revision02
In this paper the authors report a study on driving drowsiness. Specifically, they test whether a “Man-machine Response Mod” (MRM) system, which stimulates the driver by posing arithmetical problems, can alleviate driving drowsiness. The effects are assessed by a number of different measures (including EEG recordings, self-report and eye movements). All these measures seem to agree in showing a decreased drowsiness when driving was done with the MRM system, compared to a control condition. While the experiment is potentially interesting, some important details are unclear or not reported, which makes it difficult for the reader to assess the soundness of the results and their implications (see below). For example, the reporting of statistical tests is incomplete. Additionally, I think the MRM system proposed and the experimental protocol used may have some shortcomings, which are not discussed in the current version of the article. Finally, the text contains some awkward wording and sentences, and could be improved for clarity (some examples and suggestions below).
* Concerning the shortcoming of the MRM, while I am convinced that continuously presenting subjects with arithmetical problems, music and other auditory stimulation might increase vigilance and prevent to some extent drowsiness, I am not convinced it is a good idea to do it while driving in real situations. I suspect that the cognitive load induced by these tasks might make the driver less attentive to the road and, for example, slow down its reaction times in case of danger (as it’s been showed by psychological research on dual-tasking). The authors state in the discussion that “Normal driving is not affected when the equipment is used to resist fatigue, and no sides effects are caused.” however this statement cannot be supported by the data, since there are no measures of driving performance. This point could be addressed, for example, by adding in the simulation some “danger” situations that needs a prompt reaction by the driver (e.g. an infraction by another car). If the reaction times to these unexpected events did not differ between MRM mode and normal driving, then the authors could suggests that the MRM mode did not interfere with driving, but at present this statement seems unwarranted. This also present a significant limit for the application of the proposed technology in real driving, however it is not discussed.
Dear sir, thank you for your professional guidance. We added the following experimental contents.
Before the start of the experiment, we need to judge whether the extra increase of the cognitive load (arithmetic operation) induced by MRM driving mode will affect the subjects' ability to cope with unexpected problems. After driving for 1 hour in MRM driving mode (or normal driving mode), each subject needs to do an emergency test on the emergency stop of the vehicle in front of him and record the reaction time.
In section 4.5 of our manuscript, we added the following contents:
In this study, we used the MRM to relieve driving fatigue caused by long-term monotonous driving. And we also studied whether the extra increase of the cognitive load (arithmetic operation) induced by MRM driving mode will affect the subjects' ability to cope with unexpected problems. The results show that the average time taken by the subjects to respond to the emergency stop of the vehicles ahead is 0.53s in MRM driving mode and 0.87s in normal driving mode, which means that the extra increase of the cognitive load in this study will not affect the subjects' ability to cope with unexpected problems. Although this method was proven to be effective in relieving long-term driving fatigue, only a simple mathematical operation was used in our work to stimulate cranial nerves to make them active. Whether more complicated arithmetic operations are more conducive to relieving or aggravatng mental fatigue has not been studied yet.
* Finally, I think that the experimental protocol might be confounded. In the MRM condition participants were continually stimulated with acoustical stimuli (numbers, cheerful music, error warnings). This aspect of the task is not matched in the control condition. Is the reduced drowsiness due to the executing the arithmetic tasks, or simply due to the auditory stimulation? Perhaps simply listening to music while driving would reduce drowsiness as much as the MRM mode (or perhaps even more?). Because the auditory stimulation was not matched, this question cannot be addressed from the current results.
Dear sir, thank you for your professional guidance. This study does have the problem you raised. The purpose of this study is only to find a simple method to relieve driving fatigue, and it does not detail the methods involved in the MRM driving mode.
Specific comments:
*1- Some aspects of the experimental protocol are unclear or not reported. For example: I assume that each participant run the MRM and normal on different days, and at page 2, lines 66-67, it is written that the experiment was divided in two “types”: My understanding was that subjects were divided in two groups, and the order of the two condition (MRM and control) counterbalanced. Yes this is not very clear and should be stated more clearly.
Dear sir, thank you for your professional guidance. The previous expression of the experiment was indeed unclear. We have made the following amendments to this part of the content. Please see section 2.1.
Additionally, the experiment was divided into two types, with the normal driving mode being arranged in the first type of experiment and the MRM mode being arranged in the second one. All subjects needed to complete the two types of driving mode experiments. It should be noted that the MRM mode of experiment can only be carried out after all subjects have completed the first type of experiment.
*2 In the next page it is stated that half-hour of sleep was arranged (pag 3 lines 92), however the time indicated within parenthesis (0:00–0:30) seems wrong and do not allow to collocate the sleep break in the experiment (which is reported to start at 13pm and ends at 16pm for each participants).
Dear sir, thank you for your careful review of our manuscript. The rest time we want to express should be (12:00–12:30 pm). The subjects were given half an hour's sleep before the experiment to eliminate the interference to fatigue driving caused by a lack of sleep (not taking lunch break).
*I found that the description of the MRM system itself (page 4 lines 117-126) lacks details. If I understood correctly, the participant were asked to whether the sum of two numbers exceeded 100 or not, and then say whether the sum was divisible by 3. How were the digits generated? What was their range of possible values? Did the system formulated an explicit question, or simply presented the two numbers? Were the participants asked to complete both responses within 5 seconds, or were they allowed 5 second for each response? Did the system continuously proposed questions, without any breaks, or was it interleaved with pauses? What sounds were used as cheerful music and error warning signal?
Thank you very much for your constructive comments. We have made the following modifications in our manuscript. Please see section 2.2.
Firstly, the MCU randomly gives two two-digit numbers through the speaker, which take values from 0 to 100. After hearing these two numbers, the subjects need to calculate the sum of the two numbers and judge whether the sum is three digits. If this sum exceeds three digits, the subjects press the “right” answer switch, otherwise press the “left ”answer switch. Secondly, the subjects need to judge whether this sum is divisible by 3. Similarly, subjects need to press the “right” answer switch on the steering wheel if the sum can be divided by 3, otherwise press the “left” answer switch on the steering wheel. The subjects need to complete both responses within 5 seconds; otherwise, it is regarded as a mistake. Thirdly, the MCU plays cheerful music (Bandari Light Music) for 3 seconds through the speaker if the response is correct; otherwise, the speaker plays an error warning sounds (Siren Vintage Sounds). In the experiment, the MCU continuously raises questions. This fatigue relieving operation mode (MRM) was repeatedly executed until the end of the driving experiment in MRM mode. In the experiment, the data acquisition needed to stop responding to eliminate interference.
*3- Figure 4: in the text is written that the figure represents the “contrast” C (actually the clustering coefficient) at the two stages for each subject? However the figure seems to show the coefficient C averaged over subjects in the stage 1 of driving compared to the other stages (e.g. panel 1 is stages 1 vs 2, panels 2 is stage 1 vs 3, etc.). There is an inconsistency between what is written in the text and the figure.
Dear sir, thank you for your careful review of our manuscript. There is nothing wrong with what is shown in Figure 6 (the original label is “Figure.4”), but our words are not accurate. This part of the text is amended as follows:
Figure 6 (the original label is “Figure.4”) shows the comparison of the coefficient C averaged over subjects in the driving stage 1 with that of other stages (driving stage2-7). Please see lines 266-269 (page 9).
*4- Reporting of statistics: please report the statistic (e.g. t of F values) together with the p-value. Also, report the actual value of the p-value, not simply whether is exceed or not the conventional alpha threshold.
Dear sir, thank you for your careful review of our manuscript. We have made a new statistical analysis of the data and added relevant contents.
*- It is not clear exactly what test has been used to assess the differences in network characteristics (page 8).
Dear sir, thank you for your careful review of our manuscript. In this part, we use t-test, which is mentioned in section 2.3.1.We have made a new statistical analysis of the data and added relevant contents.
*For example at lines 257-258 one unique test is reported to indicate that “obviously” there was no difference in the brain network in the first three stages (please remove the word “obviously” from the text; if it was obvious then there should be no need to assess it with a statistical test). Moreover, it is reported that “p<0.05” even though the text indicate that there was no difference, so there is an inconsistency between the p-value and the text.
Dear sir, thank you for your careful review of our manuscript.We are ashamed of making such a low-level mistake. Such a low-level statement error may cause all our research results to be questioned. We have already revised this part of the content. Please see section 3.3.1(page9-10).
*- power spectrum ratio (and figure 7). Why the figure only shows data from the parietal electrode P3 and P4? Was there a difference only in these electrodes? Was the difference significant? (No statistical test is reported here).
Dear sir, thank you for your professional guidance. Previous study has shown that the brain fatigue characteristics can be easily detected from the posterior (P3, P4) brain region using EEG signals [7]. Therefore, the leads associated with the brain region can be used as the preferred ones for the analysis of driving fatigue in this study. Please see lines 41-44 (page2).
For Figure.9 (the original label is “Figure.7”), the following statistical analysis has been added. Please see lines 323-324.
The difference between the two driving modes is significant (P3: |t|=2.527>t0.05,12=2.179, p=0.03<0.05; P4: |t|=2.633>t0.05,12=2.179, p=0.02<0.05). This means that the driving fatigue of subjects in the MRM mode decreased more slowly compared with that of the normal mode.
[7] Gruzelier J H. EEG-neurofeedback for optimising performance. I: a review of cognitive and affective outcome in healthy participants[J]. Neuroscience & Biobehavioral Reviews, 2014, 44: 124-141.
*5- analysis of eye movement. It is difficult to assess the formula used (formula 8) because the sampling rate of the signal is not reported, so it is not clear to what temporal interval the 20 samples correspond to.
Dear sir, thank you for your careful review of our manuscript. In our experiment, the sampling rate of the device (Neuroscan) is selected to be 1000HZ. We added relevant contents in section 2.2
*6- The text is in several instance redundant (e.g. pag 2, lines 71-74 with pag. 3 lines 104-109), and also the paragraph at page 14 (from lines 406) is almost identical to a paragraph in the introduction (page 2 from line 43), with the exception that the reference numbers are now wrong (e.g. Zhao et al, at line 410, is reference number 19, not 20).
Dear sir, thank you for your careful review of our manuscript. We are ashamed of making such a mistake. We have already deleted the redundant content.
*7- line 452: “division”, actually is both addition and division I should disclose that I am not a native speaker, however I couldn’t help but notice that the text contain some awkward phrasing and could be improved for clarity in several places.
Dear sir, thank you for your professional guidance. Is it appropriate to replace “division” with “mathematical ”? Please see lines 469, Page 15. And the unreasonable words mentioned are revised as follows:
Here are some suggestions:
*1- “standard values of deviation” (page 5) is usually referred to simply as “standard deviation”
Thank you for your professional guidance. We have already made amendments. At the same time, we have checked the content of the full text. Please see lines 173-176, Page 5-6.
*2- pag 7, lines 202: “contain the main information about the brain” sounds a bit awkward and should perhaps be replaced with something like “are particularly useful for characterizing brain activity”
Thank you for your professional guidance. We have already made amendments. Please see line 212, Page 7.
*3- pag 7, line 224, “above researchers” should be “above studies”
Thank you for your professional guidance. We have already made amendments. Please see line 260, Page 9.
*4- pag 7, line 231, “well-founded speculation” seems an oxymoron and should perhaps be replaced with “educated guess” or something similar
Thank you for your professional guidance. We have already made amendments. Please see line267, Page 9.
*5 - line 280, “brain nerve activity” should just be “brain activity”
Thank you for your professional guidance. We have already made amendments. At the same time, we have checked the content of the full text. Please see lines 320, Page 11.
*6- pag 10, lines 188 (288?), “Studies have shown that the two eyeballs move synchronously to obtain information when human beings recognize information form the outside world” is awkward phrasing. Something like “Humans make coordinated movement of the eyes to orient the high-acuity part of the retinal on relevant objects in the world and facilitate their processing” or something along these lines might be better.
Dear sir, thank you for your professional guidance. We have already made amendments. Please see lines 328-329, Page 12.
*7- line 327, maybe “prevent” should replace “resist” here
Dear sir, thank you for your professional guidance. We refer to your suggestion in question 8 below. Is it more appropriate to replace “resist” with “alleviate”? Please see line237, Page 8.
*8- line 404, “inhibiting the deepening speed of mental fatigue” also sounds awkward. Maybe could be replaced with “alleviate the symptoms of mental fatigue” or “prevent the accumulation of mental fatigue”
Dear sir, thank you for your professional guidance. We have already made amendments. Please see line 419, Page 14.

Reviewer 3 Report
This paper proposes a detection method of driver fatigue relief, especially an evaluation algorithm using EEG, and investigates an influence of MRM(Man-machine Response Mod) on relieving driver fatigue during a long-term driving.
I suggest that the results of "subjective questionnaire" and "response error rate of subjects" should be presented first and then you introduce the findings obtained from "brain network", "eye movement" and "relative power spectrum ratio". This is because the first two indices are traditional and confirmed assessment methods and the lattes are your proposed algorithm.
You mentioned that the ratio (β/θ+α) will decrease with the deepening of mental fatigue at the section 4.2 (p.13). But, Fig. 7 suggests the ratio of "MRM" is lower than that of "Normal". Is that an inconsistent result?
Please modify the paper title to include introducing a new assessment method / algorithm using brain activities and eye movements, in addition to proposing MRM.
Author Response
Revision03
This paper proposes a detection method of driver fatigue relief, especially an evaluation algorithm using EEG, and investigates an influence of MRM(Man-machine Response Mod) on relieving driver fatigue during a long-term driving.
*1 I suggest that the results of "subjective questionnaire" and "response error rate of subjects" should be presented first and then you introduce the findings obtained from "brain network", "eye movement" and "relative power spectrum ratio". This is because the first two indices are traditional and confirmed assessment methods and the lattes are your proposed algorithm.
Dear sir, thank you for your careful review of our manuscript. We have consulted your suggestions and revised the order of these sections of the manuscript.
*2 You mentioned that the ratio (β/θ+α) will decrease with the deepening of mental fatigue at the section 4.2 (p.13). But, Fig. 7 suggests the ratio of "MRM" is lower than that of "Normal". Is that an inconsistent result?
Thank you for your professional guidance. We are sorry for our carelessness. We re-analyzed the experimental data and confirmed that the experimental results are consistent with what is described here. However, when editing the graph in Figure 9 (the original label is “Figure 7”), one coauthor mistakenly confused the two labels. We apologize for such low-level negligence. We have modified Figure 9 (the original label is “Figure 7”) , please see lines 315-318, Page 11. Please help us review it again. Thank you!
*3 Please modify the paper title to include introducing a new assessment method / algorithm using brain activities and eye movements, in addition to proposing MRM.
Dear sir, thank you for your careful review of our manuscript. The original title ofour manuscript is indeed not comprehensive enough. We have changed the title to “Study on the Effect of Man-machine Response Mode to Relieve Driving Fatigue Based on EEG and EOG”.

Round 2
Reviewer 2 Report
The authors replied adequately to almost all my comments, have followed my suggestions in improving the wording; overall I find that the manuscript has laregely improved since the first submission. There remains a few points that requires clarification.
- "Response to the emergency stop" (line 461). In the new version the authors state that they have actually performed a test to measure the effect of increased cognitive load on driving ability. This seems quite relevant, so I am a bit surprised it was not reported in the first version of the manuscript. Nonetheless, the description of the emergency test lacks details, so it is diffult for the reader to evaluate it. For example at line 130 it is stated that the test was done before the experiment? Was it done on the same set of subjects, or on a different group? (Was it on the same day?) Interestingly, it seems that the response time was faster in MRM mode. Was the difference statistically significant? What was the standard deviation of the response times across participants in this test? (Was the variance of response times across participants similar across the MRM and natural mores?)
- Statistical analyses (line 143): it is stated that ANOVA and Tukey tests have been used, however it seems that in the paper only t-tests are reported. Thus this section should be modified accordingly, by reporting that differences between normal and MRM mode were assessed using two-tailed t-tests.
Author Response
Dear sir,
Thank you for your comments and suggestions for our manuscript. I have modified the manuscript accordingly, and the detailed corrections are listed below point by point:
"Response to the emergency stop" (line 461). In the new version the authors state that they have actually performed a test to measure the effect of increased cognitive load on driving ability. This seems quite relevant, so I am a bit surprised it was not reported in the first version of the manuscript. Nonetheless, the description of the emergency test lacks details, so it is diffult for the reader to evaluate it.
Dear sir, thank you for your professional guidance. Research shows that different cognitive tasks have different effects on people's mental fatigue. Our experimental study just involves the effect of cognitive tasks on driver fatigue. We must first consider that the cognitive tasks we choose need to combat driving fatigue rather than aggravate driving fatigue. Otherwise, we may have finished the whole experiment, but the final result is meaningless. For our experiment, we have considered the effect of cognitive tasks on driving fatigue before the experiment.
*For example at line 130 it is stated that the test was done before the experiment? Was it done on the same set of subjects, or on a different group? (Was it on the same day?)
Dear sir, thank you for your professional guidance. We have made the following amendments to this part.
For 12 subjects, each of them needs to drive for 1 hour in the two driving modes (MRM driving mode and normal driving mode) respectively. When the driving time reaches 1 hour in one driving mode, the subjects need to do an emergency test on the emergency stop of the vehicle in front of him and record the reaction time. The emergency test in MRM driving mode can only be carried out after all the subjects have completed the test in the normal driving mode. And the test period is from 13:00 pm to 16:00 pm. Please see section 2.2 (page4, lines 132-139).
*Interestingly, it seems that the response time was faster in MRM mode. Was the difference statistically significant? What was the standard deviation of the response times across participants in this test? (Was the variance of response times across participants similar across the MRM and natural mores?)
Dear sir, thank you for your professional guidance. We have made the following amendments to this part.
In this study, we used the MRM to relieve driving fatigue caused by long-term monotonous driving. And we also studied whether the extra increase of the cognitive load (arithmetic operation) induced by MRM driving mode will affect the subjects' ability to cope with unexpected problems. The results show that the average time taken by the subjects to respond to the emergency stop of the vehicles ahead is 0.53s in MRM driving mode and 0.87s in normal driving mode. Moreover, the standard deviation of reaction time is 0.09s in MRM driving mode and 0.16s in normal driving mode. There was significant difference in the emergency response time between the two driving modes (|t|=6.108>t0.05,22=2.074, p=0.00<0.05). It means that the extra increase of the cognitive load in this study will not affect the subjects' ability to cope with unexpected problems. Please see section 4.5 (page15, lines 469-472).
2- Statistical analyses (line 143): it is stated that ANOVA and Tukey tests have been used, however it seems that in the paper only t-tests are reported. Thus this section should be modified accordingly, by reporting that differences between normal and MRM mode were assessed using two-tailed t-tests.
Dear sir, thank you for your careful review of our manuscript. We are ashamed of making such a low-level mistake. We have already modified this part of the content. Please see section 2.3.1 (page5, lines 149-152).
